# Development and initial validation of the Japanese healthy work environment assessment tool for critical care settings

**Mio Kitayama**[1]*, **Takeshi Unoki**[2], **Yui Matsuda**[3], **Yujiro Matsuishi**[4], **Yusuke Kawai**[5], **Yasuo Iida**[6], **Mio Teramoto**[7], **Junko Tatsuno**[8], **Miya Hamamoto**[9]

**1** Nursing Department, Kanazawa Medical University Hospital, Uchinada, Japan, **2** School of Nursing, Sapporo City University, Sapporo, Japan, **3** School of Nursing and Health Studies, University of Miami, Coral Gables, Florida, United States of America, **4** Neuroscience Nursing, Sei Roka Kango Daigaku, Tokyo, Japan, **5** Department of Nursing, Fujita Health University Hospital, Toyoake, Japan, **6** Division of General Education, Natural Sciences, Mathematics, Kanazawa Medical University, Uchinada, Japan, **7** Teachers College, Columbia University, New York, NY, United States of America, **8** Nursing Department, Kokura Memorial Hospital, Kokura, Japan, **9** Nursing Department, Tosei General Hospital, Seto, Japan

* mimio-f@kanazawa-med.ac.jp

## Abstract

### Aim

This study aims to translate the Healthy Work Environment Assessment Tool (HWE-AT) into Japanese and evaluate its validity and reliability.

### Design and methods

The authors followed the guidelines for scale translation, adaptation, and validation in cross-cultural healthcare research. After translation and back-translation, a series of pilot studies were conducted to assess comprehensibility. Subsequently, an expert panel established the content validity. Content validity was calculated using the content validity index (CVI). Finally, we verified the construct validity and calculated the test-retest reliability.

### Results

The updated HWE-AT achieved sufficient comprehensibility after conducting the two pilot tests. Content validity was calculated using the scale-level CVI/average and all the items were 1.00. The content validity indices CFI and RMSEA were 0.918 and 0.082, respectively. Intraclass correlation coefficients for all dimensions ranged from 0.618 to 0.903, indicating acceptable test-retest reliability. Our findings suggest that the Japanese version of the HWE-AT has good validity and reliability.

## Introduction

A healthy work environment is essential for providing high-quality care to patients. The work environment includes physical and psychosocial conditions that influence employee motivation, productivity, engagement, and collaboration with other employees [1]. According to the

**Data Availability Statement:** All relevant data are within the paper and its Supporting Information files.

**Funding:** The authors received no specific funding for this work.

**Competing interests:** The authors have declared that no competing interests exist.

World Health Organization, a healthy workplace is one in which everyone works together to achieve a shared vision for the health and well-being of workers and the surrounding community. It also provides all workforce members with physical, psychological, social, and organizational conditions that protect and promote their health and safety [2]. In fact, some studies have reported that a poor work environment is associated with the provision of low-quality care [3, 4]. In turn, a healthy work environment, such as an environment with higher perceptions of authentic leadership, was associated with lower burnout and higher compassion satisfaction [5]. Effectively promoting a healthy work environment (HWE) has not only prevented burnout among nurses but also decreased medication errors and pressure ulcers [6, 7].

Furthermore, there has been a growing interest in improving the work environment during the COVID-19 pandemic; however, the lack of human and personal protective equipment during the pandemic makes it difficult to maintain a safe work environment, such by as reducing the risk of infection for nurses [8]. Even then, open-ended communication, especially in leadership, and a supportive work environment could increase resilience in workers during and after the pandemic [9, 10]. In addition, by improving the work environment, nurses may be able to maintain a positive outlook even during the temporary crisis of the pandemic [11].

The American Association of Critical-Care Nurses (AACN) recognizes that a healthy work environment (HWE) is integral for nurses to contribute optimally to patient care. Moreover, a healthy work environment is vital to every healthcare team member, and respect and care for others in the healthcare team are critical [12]. To build a healthy workplace, clear standards are required. The AACN has thus developed six standards to achieve an HWE: appropriate staffing, authentic leadership, effective decision making, meaningful recognition, skilled communication, and true collaboration [13]. In 2009, the AACN developed a web-based Healthy Work Environment Assessment Tool (HWE-AT) with 18 questions on these six standards; each item is rated on a five-point Likert scale, ranging from 1 (strongly disagree with the statement) to 5 (strongly agree with the statement) [14]. As for the score cutoff, the HWE-AT score guidelines determine that 1.00–2.99 is "Needs Improvement", 3.00–3.99 is "Good", and 4.00–5.00 is "Excellent" [15]. Previous studies have indicated that the HWE-AT has good validity and reliability in a critical care setting [16]. Additionally, this tool was validated not only for nurses but also for other healthcare professionals. Therefore, all professionals in the hospital should be able to use HWE-AT to evaluate HWE.

The AACN's HWE-AT has been used to assess and improve work environments that motivate healthcare professionals to provide high-quality care, especially in the intensive care unit (ICU) in English-speaking countries [14] However, the Japanese HWE-AT has not been translated officially yet. Thus, this study aims to translate the HWE-AT into Japanese and evaluate its validity and reliability.

## Methods

In this study, the Healthy Work Environment Assessment Tool was first translated into Japanese and validated for content validity. Further, it assessed for construct validity, reliability, and internal consistency.

### Summary of the Japanese HWE-AT development

Illustrates the steps taken by the authors to develop the Japanese HWE-AT (Fig 1).

### Translation process and content validity study

**Translation procedure.** The translation was initiated after obtaining permission from the AACN. To develop the translation, we followed Sousa's guidelines [17]. Primarily, the original

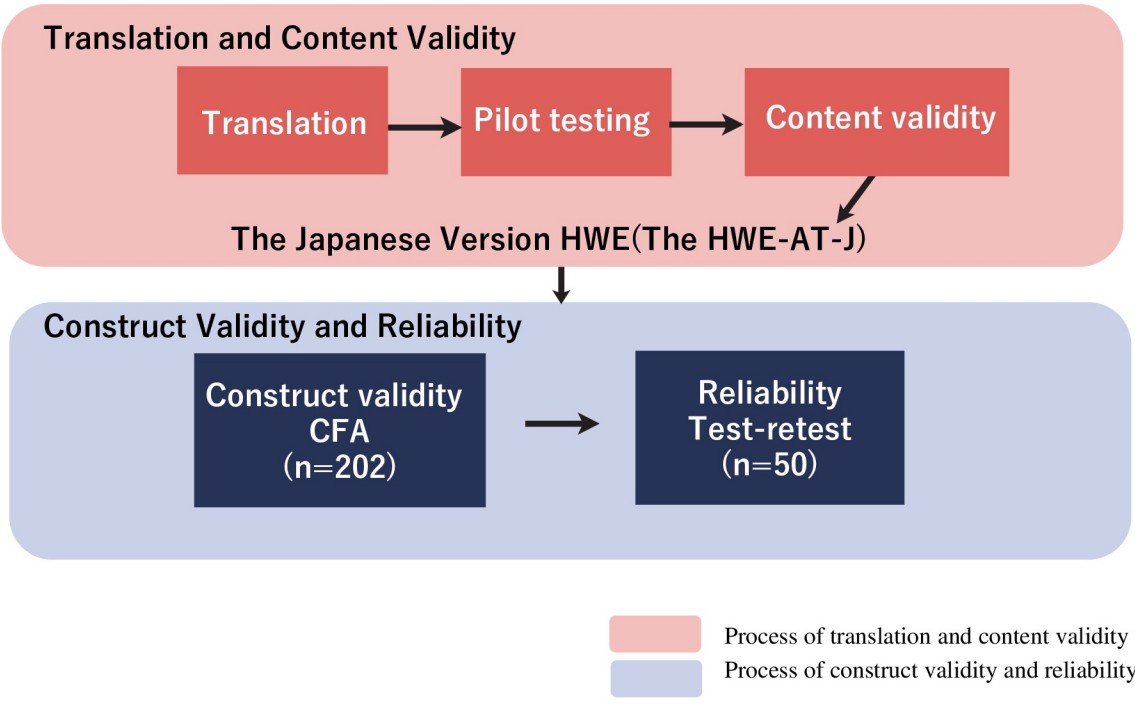

**Fig 1. Summary of the HWE-AT-J development.** First, the HWE-AT-J was developed through translation and content validation. And then construct validity and reliability were conducted.

version of the HWE-AT was translated independently into Japanese by two bilingual nurses. One was a nurse educator who formerly lived in the U.S. as a postdoctoral fellow and had worked as a nurse manager in the ICU in Japan for 25 years. The other nurse has worked in the ICU settings for 16 years and has been registered as a certified nurse in intensive care by the Japanese Nursing Association. The two translated versions were integrated into one Japanese version of the HWE-AT (hereafter referred to as "draft version"), based on the discussion among the translation team. This team consists of a group of bilingual nurses: a nurse educator, another nurse educator working in the United States, two nurse managers, three ICU nurses, and a graduate student in the U.S. who had previously worked as a nurse in Japan. Subsequently, the integrated document was back-translated into English by a professional native English-speaking translator who was unfamiliar with HWE-AT. Subsequently, the original and back-translated versions were compared. Following this process, the translation team finalized a preliminary Japanese HWE-AT based on the draft version.

**Pilot testing for clarification of the translated tools.** A series of pilot studies were conducted to evaluate the clarity and validity of each item in the preliminary Japanese HWE-AT. We invited participants who were working as ICU nurses via one of the mailing lists of the Japanese Society of Education for Physicians and Trainees in Intensive Care. Forty nurses voluntarily participated in a web-based survey. Each participant was asked to rate the instructions for the scale and each item using a dichotomous answer option (clear or unclear). If more than 20% of participants rated an instruction/item as unclear, we modified it to improve clarity and understanding. We repeated these steps until most participants rated all the items as clear (80% or more rated the items as clear). In case of repeated tests, the same participant was not

allowed to answer the survey again. Therefore, a pre-final version of the Japanese HWE-AT was developed.

**Content validity testing for expert panel.** The content validity of the pre-final Japanese HWE-AT was evaluated by 10 experts who were knowledgeable about this content and experienced in hospital settings. All of these experts have more than ten years of experience as critical care nurses. The expert panel was asked to assess each item of the instrument for content equivalence concerning how it relates to the HWE using the following scale: 1 = "not relevant," 2 = "unable to assess its relevance," 3 = "relevant but needs minor alteration," and 4 = "very relevant and succinct." We also requested that these 10 experts write any comments necessary about the clarity of the items and provide suggestions and recommendations to improve their formulation.

The item-level content validity index (CVI) was then calculated. The scale was dichotomized by combining answer options 3 and 4 to calculate the scale-level CVI/average (S-CVI/Ave). We then calculated the ratio by dividing the CVI/average by the total number of experts [18]. Subsequently, we developed the Japanese Healthy Work Environment Assessment Tool (HWE-AT-J).

## Construct validity testing

**Participant selection for construct validity.** The authors collected additional data to establish construct validity. The eligible participants were nurses working in ICUs in Japan. A convenience sampling method was used, and the participants were recruited through the mailing lists of the Japanese Society of Intensive Care Medicine and the Japanese Society of Education for Physicians and Trainees in Intensive Care. We asked participants who had already completed the questionnaire to distribute the invitation to other local mailing lists or social networking sites. Data were collected between October and December of 2020. The questionnaire was web-based and anonymous. Each participant was asked to complete the HWE-AT-J and provide demographic characteristics, including gender, years of ICU experience, years of nursing experience, type of hospital, working unit, position, and qualification.

**Construct validity testing.** Confirmatory factor analysis (CFA) was performed to assess the theoretical connectedness and structural equivalence of the original and translated HWE-AT. CFA is a typical method for assessing the relationship between a factor and an observed variable based on a prior measure.

**Reliability testing.** *Participant selection for reliability*. We recruited participants using the same method utilized to evaluate construct validity. We sent messages to those who provided consent to be contacted for the test and retest surveys, collecting basic data characteristics and e-mail addresses.

**Reliability testing.** We evaluated the reliability of the HWE-AT-J using the test-retest method. The participants were asked to complete the HWE-J. Two weeks after they answered the HWE-J (test), we asked participants to complete the HWE-AT-J again (retest). The reliability of the Japanese translated instrument was determined using Cronbach's alpha as a measure of internal consistency of the items within the six domains of the HWE-AT-J. To estimate Cronbach's alpha, we used a larger of the two cohorts to avoid statistical errors.

**Sample size calculation for validity and reliability testing.** Adequate statistical power contributes to observing authentic relationships in a dataset [19]. Minimum sample sizes in absolute numbers were the first rule of thumb, suggesting that any $N > 200$ offers adequate statistical power for data analysis [20]. Therefore, we set the sample size of the confirmatory factor analysis (CFA) to 200 and recruited participants. We also used Kaiser-Meyer-Olkin test to confirm that the sample size was appropriate [21].

For the reliability test, as an indicator of test-retest reliability, we used the two-way random-effects model of the intraclass correlation coefficient (ICC). We calculated the sample size of the ICC statistics based on the formula recommended by Zou [22] and set the null hypothesis as 0.6, alternative hypothesis = 0.8, alpha = 0.5, and test power = 0.8. Based on these numbers, 49 patients were required. Therefore, we selected 50 participants.

**Data analysis.**   Participant characteristics were expressed as percentages and numbers, medians, and interquartile ranges (IQR) for non-normally distributed data or means and standard deviations (SD) for normally distributed data. Construct validity was evaluated using the Confirmatory Factor Analysis (CFA). We did not perform exploratory factor analysis (EFA) because as far as we know the HWE—AT was not developed using EFA [23]. Moreover, the purpose of this study was to consider structural equivalence between the Japanese version and the original version. Therefore, we chose to perform CFA. Under the CFA, comparative fit index (CFI), and root-mean-square error of approximation (RMSEA), two frequently used models, were adopted in this study [24]. CFI ranges between 0 and 1 and is generally appropriate at values of 0.90 [25]. RMSEA needs a cut-off value to be changed depending on the sample size, and in our study, we set the cut-off value to 0.1 [26]. We evaluated ceiling and floor effects for each question item. Additionally, we assessed the ceiling and floor effects corresponding to the percentage of respondents who obtained minimum (1) or maximum (5) for each question. Above 15% of respondents either minimum or maximum indicated there was a problem with validity [27].

We used the two-way random-effects model of the intraclass correlation coefficient (ICC) to measure test-retest reliability. ICCs < 0.5 indicate poor reliability; ICCs between 0.5 and 0.75 indicate moderate reliability; ICCs between 0.75 and 0.9 indicate good reliability; and ICCs > 0.9 indicate excellent reliability [28]. We also calculated Cronbach's alpha as an internal consistency measure to ensure that the subscale questions measured similar concepts. Cronbach's α of 0.7 to 0.8 is generally considered good [29].

**Ethical considerations.**   Ethical approval for the research protocol was granted by the ethical review board of the Kanazawa Medical University Hospital, Ishikawa, Japan, (approval ID H275). For participant consent, an instruction sheet was attached on the web and a check box was provided so that participants could indicate their willingness to participate. The consent form had to be viewed and checked to be able to respond. In addition, contact information was attached so that participants could request verbal explanations or written consent, and this was handled on an individual basis.

## Results

### Translation and content validity

**Pilot testing.**   In the first test, five items in the HWE-AT (1, 2, 6, 7, and 10) were rated as unclear by equal to or greater than 20% of participants. Based on these results, we considered ways to clarify expressions and revise the meaning of the items to improve comprehension. In particular, we tailored the item description, noting the U.S.-Japan differences in the scope of nursing practice and the name designations of nurse managers. Regarding the different position titles between the U.S. and Japan, we received permission from the AACN to adapt the names appropriately to the Japanese medical hierarchy. After the revision, we conducted a second pilot test with 40 nurses who did not participate previous survey. Consequently, this test showed that all items were clearly described (maximum lack of clarity rate of less than 20%).

**Content validity testing.**   The expert panel evaluated the comprehensibility between the items of the original HWE-AT and the pre-final Japanese HWE-AT. Experts suggested a more appropriate expression of Japanese terms that were either not explicit or misleading. For

content validity, S-CVI/Ave was 1.00. After establishing content validity, we called the HWE-AT translated into Japanese "HWE-AT-J." as shown in S1 Table.

## Construct validity testing

**Characteristics of the participants for construct validity testing.**   The total number of participants was 202, and there were no missing values for any questionnaire item. We also evaluated the measures of sampling adequacy (MSA) using the Kaiser-Meyer-Olkin test for each item, and the results indicate that sample size was sufficient for factor analysis (meritorious:0.87~ marvelous:0.95), as shown in S2 Table.

Participant characteristics are shown in Table 1. Approximately 34% of the participants were female, and the largest number of participants had 10 to 15 years of ICU experience ($n = 65$). Nearly half of the participants worked in the university hospital settings ($n = 92$), and a majority worked in an ICU ($n = 132$). Among the participants, 61 were certified nurses, the largest number.

**Ceiling and floor effect.**   The authors determined that the ceiling effect was observed when 0% to 14% of respondents rated "5". The floor effect was determined when 7% to 40% of respondents rated as "0". We found Q3, Q 4, Q7, Q8, Q10, Q14, Q16, Q17, and Q 18 had floor effects.

**Construct validity testing.**   The mean scores and standard deviations are shown in Table 2. The indices of the CFA are listed in Table 3. The fit indices obtained were 0.918 for CFI and 0.082 for RMSEA.

## Reliability testing

**Participant characteristics.**   The total number of participants was 50, and there were no missing values for any questionnaire item. As shown in Table 1, the participants were about 30% female, and the largest number of participants had 5 to 9 years of ICU experience ($n = 19$). Fifty-eight percent of the participants worked in a university hospital setting ($n = 29$). Seventy-six percent of them worked in medical-surgical ICUs ($n = 38$). Among the participants, 15 were certified nurses, the largest number.

**Cronbach's α.**   We used the same data when we tested the construct validity to estimate Cronbach's alpha (n = 202). As indicated in Table 4, the HWE-AT-J showed adequate reliability, as estimated by Cronbach's α, for all domains: 0.607–0.811.

**Intraclass correlation coefficients.**   The ICCs are presented in Table 5. The ICCs for all dimensions ranged from 0.618 to 0.903. Therefore, it was shown to have moderate reliability.

## Discussion

The authors demonstrated the comprehensibility, validity, and reliability of the HWE-AT-J. We used Sousa's translation guidelines [17] since we believe that the quality of the data obtained from the translated scale depends on the accuracy of the translation. We paid particular attention to the differences between the U.S. and Japanese healthcare systems and hierarchies within nursing positions. Differences in job titles between the two countries were particularly problematic. For example, the word "leader" in the U.S. indicates managers, certified nurse leaders, and advanced registered nurse practitioners. However, in Japan, it implies nurses in charge of a shift (equivalent to "charge nurses" in the U.S.). Therefore, we defined and included the titles of Japanese nurse leaders in hospitals, such as "chief nurses," "certified nurses," and "nurse managers." A Certified nurse (CN) is those who received at least 600 hours of training in a special field and who have often served as leaders of a unit because of their specialized knowledge and expertise. Also, a Certified Nurse Specialist (CNS) has

**Table 1. Participant characteristics for construct validity and reliability testing.**

| Characteristic | Construct Validity Testing n = 202 | [c]Reliability Testing n = 50 |
|---|---|---|
| | n (%) | n (%) |
| Gender | | |
| Male | 134 (66) | 35 (70) |
| Female (%) | 68 (34) | 15(30) |
| Years of ICU experience | | |
| <5 | 41 (20) | 8 (16) |
| 5–9 | 61 (30) | 19 (38) |
| 10–15 | 65 (32) | 16 (32) |
| 16–20 | 26 (13) | 6 (12) |
| >20 | 9 (5) | 1 (2) |
| Years of Nursing experience | | |
| <5 | 14 (7) | - |
| 5–9 | 48 (24) | - |
| 10–15 | 58 (29) | - |
| 16–20 | 44 (21.5) | - |
| >20 | 38 (18.5) | - |
| Working unit | | |
| [a]ICU | 132 (65.5) | 38 (76) |
| Emergency ICU | 38 (19) | 7 (14) |
| [b]CCU | 27 (13) | 5 (10) |
| ICU/CCU | 1 (0.5) | 0 (0) |
| Pediatric ICU | 2 (1) | 0 (0) |
| Stroke Care Unit | 1 (0.5) | 0 (0) |
| Surgical ICU | 1 (0.5) | 0 (0) |
| Hospital facilities | | |
| University hospital | 92 (46) | 29 (58) |
| Public hospital | 35 (17) | 7 (14) |
| National hospital | 12 (6) | 13 (26) |
| Private hospital | 63 (31) | 1 (2) |
| Position | | |
| Staff | 147 (73) | 31 (62) |
| Administrator | 55 (27) | 19 (38) |
| Qualification | | |
| Registered Nurse | 99 (49) | 31 (62) |
| Nurse Practitioner | 3 (1) | 0(0) |
| Certified Nurse Specialist | 22 (11) | 3 (6) |
| Certified nurse | 61 (30) | 15 (30) |
| Others | 17 (8) | 1 (2) |

*Note.*

[a]ICU = Intensive Care Unit

[b]CCU = Cardiac Care Unit

[c]Reliability Testing = It does not collect information on years of nursing experience in reliability testing.

completed a master's degree in a specific field. One research team member who is a registered nurse and had previously worked in an ICU in the U.S. helped clarify these differences.

 Floor effects were observed in the following items (Q3, Q4, Q7, Q8, Q10, Q14, Q16, Q17 and Q18). As consistent with the original scale, a five-point Likert scale was used to measure

**Table 2. Means, standard deviations for each questionnaire and factor loading.**

| | Mean | [a] SD | [b]Factor loading |
|---|---|---|---|
| Skilled communication | | | |
| Q.1: Maintain frequent communication | 3.28 | 0.921 | 0.63 |
| Q.6: Input seeking for decision-making | 2.94 | 0.941 | 0.73 |
| Q.14: Staff members let people know when they've done a good job | 2.87 | 1.019 | 0.66 |
| True collaboration | | | |
| Q.2: Actions match words | 3.18 | 0.929 | 0.77 |
| Q.10: Enough staff to maintain patient safety | 2.51 | 1.004 | 0.67 |
| Q.15: Motivating opportunities for personal growth | 3.02 | 0.972 | 0.82 |
| Effective decision-making | | | |
| Q.7: Consistent use of data-driven, logical decision-making process | 2.75 | 0.951 | 0.80 |
| Q.11: Right mix of nurses and other staff to ensure optimal outcomes | 3.17 | 0.931 | 0.69 |
| Q.16: Staff have positive relationship with nurse leaders | 3.44 | 0.908 | 0.74 |
| Appropriate staffing | | | |
| Q.3: Zero tolerance for disrespect and abuse | 3.10 | 1.067 | 0.78 |
| Q.8: Right departments, professions, groups are involved | 2.97 | 1.004 | 0.82 |
| Q.12: Support services level allows nurses and staff to focus on care | 3.05 | 0.971 | 0.68 |
| Meaningful recognition | | | |
| Q.4: Staff involved in decision-making | 2.50 | 1.052 | 0.75 |
| Q.9: Patient's perspective is considered in important decisions | 3.05 | 0.976 | 0.78 |
| Q.17: Nurse leaders understand dynamics at point of care | 2.94 | 1.016 | 0.78 |
| Authentic leadership | | | |
| Q.5: Able to influence policies, procedures, and bureaucracy | 3.16 | 0.844 | 0.67 |
| Q.13: Formal recognition system makes staff feel valued | 3.01 | 0.980 | 0.82 |
| Q.18: Nurse leaders play role in making key decisions | 3.01 | 1.005 | 0.71 |

Note

[a]SD = Standard Deviations

Note

[b]Factor loading is standardized.

**Table 3. CFA fit indices[a].**

| CFA fit indices | |
|---|---|
| **Factor** | **Indices** |
| $\chi^2$ | 233 |
| df | 12 |
| | p<0.01 |
| [b]CFI | 0.918 |
| [c]RMSEA | 0.082 |

Note

[a]CFA = Confirmatory Factor Analysis

Note

[b]CFI = comparative fit index

Note

[c]RMSEA = root-mean-square error of approximation.

**Table 4. Reliability testing: Cronbach's α.**

| | Scale mean that items are deleted | Variance of frequencies that an item is deleted | Corrected items Total Correlation | Cronbach's alpha for the case when an item is deleted |
|---|---|---|---|---|
| Domain Skilled Communications (Over all Cronbach's α:0.707) | | | | |
| Q1 | 5.81 | 2.823 | 0.507 | 0.637 |
| Q6 | 6.15 | 2.585 | 0.585 | 0.541 |
| Q14 | 6.21 | 2.6.6 | 0.486 | 0.669 |
| Domain True Collaboration (Over all Cronbach's α:0.754) | | | | |
| Q2 | 5.53 | 2.847 | 0.621 | 0.629 |
| Q10 | 6.20 | 2.966 | 0.488 | 0.780 |
| Q15 | 5.69 | 2.662 | 0.647 | 0.594 |
| Domain Effective Decision Making (Over all Cronbach's α:0.786) | | | | |
| Q7 | 6.60 | 2.658 | 0.611 | 0.728 |
| Q11 | 6.19 | 2.751 | 0.595 | 0.744 |
| Q16 | 5.92 | 2.640 | 0.674 | 0.659 |
| Domain Appropriate Staffing (Over all Cronbach's α:0.799) | | | | |
| Q3 | 6.01 | 2.980 | 0.677 | 0.690 |
| Q8 | 6.15 | 3.145 | 0.690 | 0.677 |
| Q12 | 6.07 | 3.577 | 0.570 | 0.790 |
| Domain Meaningful Recognition (Over all Cronbach's α:0.811) | | | | |
| Q4 | 5.99 | 3.174 | 0.653 | 0.750 |
| Q9 | 5.44 | 3.411 | 0.656 | 0.746 |
| Q17 | 5.54 | 3.234 | 0.674 | 0.726 |
| Domain Authentic Leadership (Over all Cronbach's α:0.778) | | | | |
| Q5 | 6.03 | 3.024 | 0.622 | 0.697 |
| Q13 | 6.17 | 2.661 | 0.655 | 0.654 |
| Q18 | 6.17 | 2.781 | 0.574 | 0.748 |

the items. The result reflects the current situation of Japanese nursing practice that is hierarchical in nature. Therefore, the floor effect does not hamper the generalization of the scale adequacy.

The overall CVI score was excellent. Using web-based surveys as a data collection method, we obtained responses from ICU nurses with considerable variability in years of experience and roles. As a result, we believe that the data show variability in experience with diverse perceptions. In terms of validation, the model fit was moderate and considered acceptable. CFI and RMSEA were 0.918 and 0.082, respectively. A CFI value of 0.90 or higher is deemed acceptable, and a cut-off score of RMSEA is 0.1 [25, 26]. Therefore, our fit model was good. The factor loadings for each factor of HWE-AT-J showed that all items were acceptable, given the recommended cut-off value of 0.4 [30]. In a study of healthcare workers at a children's hospital in the U.S., the range of the HWE-AT's factor loading was 0.63–0.82 [23]. In other studies, the factor loading ranged from 0.485 to 0.824 [16]; therefore, we considered the HWE-AT-J factor loading more acceptable. Also, the result of sample size by using Kaiser-Meyer-Olkin test was appropriate [21].

Reliability was considered moderate and acceptable. There are no clear cut-off scores for ICC criteria, but it is generally considered that 0.61–0.80 is substantial [31]. In our study, the range was 0.618–0.903, which we believe is appropriate. Cronbach's alpha coefficients ranged from 0.620 to 0.907, with some internal consistency. In the AACN HWE-AT study, Cronbach's alpha was 0.97, indicating high internal consistency [30]. However, in previous studies,

**Table 5. Intraclass correlation coefficient.**

|  | [a]ICC | 95% CI | F test value | P value |
|---|---|---|---|---|
| Q1 | 0.762 | 0.580–0.865 | 4.156 | <0.01 |
| Q2 | 0.618 | 0.331–0.783 | 2.631 | <0.01 |
| Q3 | 0.811 | 0.666–0.893 | 5.208 | <0.01 |
| Q4 | 0.762 | 0.581–0.865 | 4.167 | <0.01 |
| Q5 | 0.756 | 0.573–0.861 | 4.146 | <0.01 |
| Q6 | 0.77 | 0.595–0.870 | 4.316 | <0.01 |
| Q7 | 0.787 | 0.622–0.88 | 4.638 | <0.01 |
| Q8 | 0.812 | 0.669–0.894 | 5.255 | <0.01 |
| Q9 | 0.83 | 0.700–0.904 | 5.785 | <0.01 |
| Q10 | 0.854 | 0.733–0.919 | 7.443 | <0.01 |
| Q11 | 0.854 | 0.744–0.917 | 7.002 | <0.01 |
| Q12 | 0.769 | 0.596–0.869 | 4.358 | <0.01 |
| Q13 | 0.71 | 0.487–0.836 | 3.401 | <0.01 |
| Q14 | 0.82 | 0.683–0.897 | 5.519 | <0.01 |
| Q15 | 0.771 | 0.596–0.870 | 4.316 | <0.01 |
| Q16 | 0.653 | 0.386–0.804 | 2.87 | <0.01 |
| Q17 | 0.903 | 0.829–0.945 | 10.699 | <0.01 |
| Q18 | 0.73 | 0.521–0.848 | 3.664 | <0.01 |

Note. ICC = Intraclass Correlation Coefficient.

they ranged from 0.75–to0.86 [29], which showed lower internal consistency when compared to our survey. Therefore, we determined that reliability was acceptable.

## Limitation

First, we generally followed Sousa's guidelines, but we could not fully follow its back-translation step due to limited human resources. Specifically, the Sousa approach recommends comparing two back translations; however, we could only conduct a single back translation. Second, we used a convenience sampling method to recruit participants for CFA, which may have led to selection bias. However, the characteristics of the participants varied in terms of their ICU experience, the proportion of participants working in a university hospital setting, and the proportion of staff to administrators. Thus, we determined that the sampling method did not significantly affect our findings.

## Clinical implications/Research implication

The HWE-AT-J provides a way to determine the health of the work environment in healthcare facilities (e.g., units, departments, hospitals). The tool will help identify and evaluate current standards. The HWE-AT score guidelines determine that 1.00–2.99 is "Needs Improvement", 3.00–3.99 is "Good", and 4.00–5.00 is "Excellent" [15]. If a unit falls below the standard, issues can be identified, and steps can be taken to resolve them. Moreover, AACN had previously surveyed HWE on a five-point scale between 2006 and 2018 [32]. Using the HWE—AT-J, we will be able to evaluate the current state in each unit.

The HWE-AT-J enables researchers to measure the health of nursing units' work environments. Future studies can measure the health of the work environment in relation to patient care quality and its associated factors for a stratified sample of ICU nurses in Japan.

## Conclusion

The Japanese version of the HWE-AT has good comprehensibility, validity, and reliability. The HWE-AT-J was the first Japanese translation of the HWE-AT, which showed promising preliminary results in creating a healthy environment in Japanese ICUs.

## Supporting information

**S1 Table. Japanese version healthy work environment assessment tool.**
(DOCX)

**S2 Table. Measures of sampling adequacy.**
(DOCX)

**S1 Data.**
(XLSX)

**S2 Data.**
(XLSX)

## Acknowledgments

We would also like to thank the expert panel for their cooperation in content validity. We would like to thank Dr. James Britton and the University of Miami School of Nursing and Health Studies for the editorial support.

## Author Contributions

**Conceptualization:** Mio Kitayama, Takeshi Unoki.

**Data curation:** Mio Kitayama, Takeshi Unoki, Yujiro Matsuishi.

**Formal analysis:** Mio Kitayama, Takeshi Unoki, Yujiro Matsuishi, Yasuo Iida.

**Funding acquisition:** Mio Kitayama.

**Investigation:** Mio Kitayama, Takeshi Unoki.

**Methodology:** Mio Kitayama, Takeshi Unoki.

**Project administration:** Mio Kitayama, Takeshi Unoki.

**Resources:** Mio Kitayama, Takeshi Unoki.

**Software:** Mio Kitayama, Takeshi Unoki, Yujiro Matsuishi.

**Supervision:** Mio Kitayama, Takeshi Unoki.

**Validation:** Mio Kitayama, Takeshi Unoki.

**Visualization:** Mio Kitayama, Takeshi Unoki.

**Writing – original draft:** Mio Kitayama.

**Writing – review & editing:** Mio Kitayama, Takeshi Unoki, Yui Matsuda, Yujiro Matsuishi, Yusuke Kawai, Mio Teramoto, Junko Tatsuno, Miya Hamamoto.

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
