## [Decision Letter · Decision Letter 0]

11 Feb 2022

PONE-D-21-39634Development and Initial Validation of the Japanese Healthy Work Environment Assessment Tool for Critical Care SettingsPLOS ONE

Dear Dr. Kitayama,

Thank you for submitting your manuscript to PLOS ONE. After careful consideration, we feel that it has merit but does not fully meet PLOS ONE’s publication criteria as it currently stands. Therefore, we invite you to submit a revised version of the manuscript that addresses the points raised during the review process.

We look forward to receiving your revised manuscript.

Kind regards,

Fatih Özden, PhD

Academic Editor

PLOS ONE

Journal Requirements:

2. During your revisions, please note that a simple title correction is required: Remove "PLOS ONE Dear Editor-in-Chief". Please ensure this is updated in the manuscript file and the online submission information.

3. Please provide additional details regarding participant consent. In the Methods section, please ensure that you have specified (1) whether consent was informed and (2) what type you obtained (for instance, written or verbal). If your study included minors, state whether you obtained consent from parents or guardians. If the need for consent was waived by the ethics committee, please include this information.

“The authors thank the Clinical Research Support Office of the Medical Research Institute at Kanazawa Medical University for their excellent support in this study.”

6. Please ensure that you refer to Figure 1 in your text as, if accepted, production will need this reference to link the reader to the figure.

Additional Editor Comments:

Dear Authors,

Thanks for your submission. After the careful consideration regarding the comments of two reviewers, my decision is a major revision. Please carefully read and fulfill all the comments. Also, provide a point to point response file.

King Regards

Reviewers' comments:

Reviewer's Responses to Questions

**Comments to the Author**

1. Is the manuscript technically sound, and do the data support the conclusions?

Reviewer #1: Yes

Reviewer #2: Partly

2. Has the statistical analysis been performed appropriately and rigorously? 

Reviewer #1: Yes

Reviewer #2: Yes

3. Have the authors made all data underlying the findings in their manuscript fully available?

Reviewer #1: Yes

Reviewer #2: No

4. Is the manuscript presented in an intelligible fashion and written in standard English?

Reviewer #1: Yes

Reviewer #2: Yes

5. Review Comments to the Author

Reviewer #1: Thank you for asking me to review this article. The results are very interesting, the article is clearly written and organized, and I believe the article will be of interest to the readers.

There are some minor fixes to consider.

Introduction

On page 5, line 89, a few more recent studies examining the impact of the work environment on patient and employee health/outcomes should be included.

The Covid-19 pandemic has been an important factor affecting intensive care environments, workforce planning, rapid decision making, communication and collaboration. This is not mentioned at all in the article. Briefly, the effect of the pandemic on the intensive care work environment can be mentioned.

Background

Page 6, line 99. Does this measurement tool (HWE-AT) have a cutoff score? How is the score obtained from the scale interpreted? It should be explained briefly.

Methods/ Content validity

What technique was content validation done (Davis?)

Page 8, line 141. It is the expert opinion experienced in the hospital setting? Or were those with intensive care experience selected?

Participant Selection for Construct Validity

Is the delivery of care in intensive care units in Japan cascaded according to the severity of the patient condition? Which level (1st, 2nd or 3rd level) of the nurses participating in the study work in the intensive care unit?

Results

Pilot testing

Were 40 nurses participating in the study included in the sample of the study at this stage? It must be disclosed.

Sample size

Page 10, line 184. Has the sample size been confirmed by the Kaiser-Meyer-Olkin test as to whether the sample size is sufficient for factor analysis?

Before the discussion, the characteristics of the final form should be given briefly. It is as if any item was removed from the Japanese scale and the sub-factor structure was preserved.

Clinical Implications/Research Implication

Page 17, lines 301-302. “standard” should be explained, whether a unit falls below the standard will be evaluated according to a written/valid procedure or according to the score obtained from the scale!

Figure 1

“Testeing” should be fixed

Reviewer #2: #1

Since HWE-AT-j is an evaluation method derived from the original HWE-AT, it is necessary to discuss the comparison with the original HWE-AT (distribution of the sample, factor loading for each question, etc.).

#2

The author started the analysis with confirmatory factor analysis. Please explain why you did not consider factor structure, factor loading, etc. in the exploratory factor analysis.

#3

The reader cannot evaluate the ceiling effect and floor effect from table2. Please make it a general table (mean, SD).

#4

In table 5, Cronbach's increases when Q10 is deleted. Please explain this in the discussion. Is it the same trend in the original paper?

#5

(line 241) There is no mention of this in table2. Please correct it.

#6

Please list χ2 and df in table3. The researcher is free to use any index to some extent, but χ2 is usually necessary.

#7

What is the reason why the factor loading in table3 is not standardized?　In other papers, standardized factor loading is used.

#8

In Japan, what is the difference between a certified nurse and a specialist?　What kind of license is it?　Please describe.

#9

The percentages for staff and administration in table 4 are blank. Please correct it.

#10

Why don't you combine the characteristics in table 1 and 4 into the same table? This is a suggestion.

#11

Please attach the Japanese version of the questionnaire (HWE) prepared by the author to the supporting information.

6. PLOS authors have the option to publish the peer review history of their article (what does this mean?). If published, this will include your full peer review and any attached files.

Reviewer #1: No

Reviewer #2: No

---

## [Author Response · Author response to Decision Letter 0]

4 Apr 2022

Response to Editor and Reviewer

Journal Requirements:

Comment

#1

Please ensure that your manuscript meets PLOS ONE's style 

requirements, including those for file naming. The PLOS ONE style 

templates can be found at

 and

Response

Thank you for pointing this out. We have made the corrections as per PLOS ONE’s style requirements.

Comment

#2

 Please provide additional details regarding participant consent. In the Methods section, please ensure that you have specified (1) whether consent was informed and (2) what type you obtained (for instance, written or verbal). If your study included minors, state whether you obtained consent from parents or guardians. If the need for consent was 

waived by the ethics committee, please include this information.

Response

 Thank you for your comment. We added these sentences.

------Revised Manuscript------

P12, Line221

For participant consent, an instruction sheet was attached on the web and a check box was provided so that participants could indicate their willingness to participate. The consent form had to be viewed and checked to be able to respond. In addition, contact information was attached so that participants could request verbal explanations or written consent, and this was handled on an individual basis.

Comment

#3

 Thank you for stating the following financial disclosure:

“The funders had no role in study design, data collection and analysis, 

decision to publish, or preparation of the manuscript.”

Response

Thank you for your comment. The authors received no specific funding for this work.

We apologize for any misleading representations. In addition, we got to advice about English expressions, so we added that part.

------Revised Manuscript------

These sentences were deleted.

P28, Line382

The authors thank the Clinical Research Support Office of the Medical Research Institute (institution name blinded for peer review) for their excellent support in this study.

These sentences were added.

P28, Line384

We would like to thank Dr. James Britton and the University of Miami School of Nursing and Health Studies for the editorial support.

Comment

#4

Please ensure that you refer to Figure 1 in your text as, if accepted, production will need this reference to link the reader to the figure.

Response

Thank you for your comment. We added these sentences.

------Revised Manuscript------

P6, Line 106

Reviewer #1

Thank you for asking me to review this article. The results are very interesting, the article is clearly written and organized, and I believe the article will be of interest to the readers. There are some minor fixes to consider.

Comment

#1

Introduction

On page 5, line 89, a few more recent studies examining the impact of the work environment on patient and employee health/outcomes should be included.

Response

Thank you for your comment. We have added a few more recent studies related to the impacts of work environment on patients and employees.

------Revised Manuscript------

P4, Line 67

In turn, a healthy work environment, such as an environment with higher perceptions of authentic leadership, was associated with lower burnout and higher compassion satisfaction [5]. Effectively promoting a healthy work environment (HWE) has not only prevented burnout among nurses but also decreased medication errors and pressure ulcers [6,7].

Comment

#2

Introduction

The Covid-19 pandemic has been an important factor affecting intensive care environments, workforce planning, rapid decision making, communication and collaboration. This is not mentioned at all in the article. Briefly, the effect of the pandemic on the intensive care work environment can be mentioned.

Response

Thank you for your comment. We have added material about the effect of the COVID-19 pandemic on work environment in intensive care units.

------Revised Manuscript------

P4, Line 72

Furthermore, there has been a growing interest in improving the work environment during the COVID-19 pandemic; however, the lack of human and personal protective equipment during the pandemic makes it difficult to maintain a safe work environment, such by as reducing the risk of infection for nurses [8]. Even then, open-ended communication, especially in leadership, and a supportive work environment could increase resilience in workers during and after the pandemic [9, 10] In addition, by improving the work environment, nurses may be able to maintain a positive outlook even during the temporary crisis of the pandemic [11]. 

Comment

#3

Background

Page 6, line 99. Does this measurement tool (HWE-AT) have a cutoff score?

How is the score obtained from the scale interpreted? It should be explained briefly.

Response

Thank you for your comment. We added a statement that the scale is a five point Likert scale and discussed the cutoff score of each question.

------Revised Manuscript------

P5, Line88

each item is rated on a five-point Likert scale, ranging from 1 (strongly disagree with the statement) to 5 (strongly agree with the statement) [14]. As for the score cutoff, the HWE-AT score guidelines determine that 1.00-2.99 is “Needs Improvement”, 3.00-3.99 is “Good”, and 4.00-5.00 is “Excellent” [15].

Comment

#4

Methods/ Content validity

What technique was content validation done (Davis?) Page 8, line 141. It is the expert opinion experienced in the hospital setting? Or were those with intensive care experienceselected?

Response

Thank you for your comment. We added the following sentence.

------Revised Manuscript------

P8, Line140

All of these experts have more than ten years of experience as critical care nurses.

Comment

#5

Participant Selection for Construct Validity

Is the delivery of care in intensive care units in Japan cascaded according to the severity of the patient condition? Which level (1st, 2nd or 3rd level) of the nurses participating in the study work in the intensive care unit?

Response

Thank you for your comment. There is no criteria dividing the ICU by severity of illness in Japan.

Comment

#6

Results

Pilot testing

Were 40 nurses participating in the study included in the sample of the study at this stage? It must be disclosed.

Response

Thank you for your comment. Forty nurses who participated in the study were not include in the sample of the study at this stage. We added the following sentence.

------Revised Manuscript------

P8, Line135

In case of repeated tests, the same participant was not allowed to answer the survey again.

Comment

#7

Sample size

Page 10, line 184. Has the sample size been confirmed by the Kaiser- Meyer-Olkin test as to whether the sample size is sufficient for factor analysis? Before the discussion, the characteristics of the final form should be given briefly. It is as if any item was removed from the Japanese scale and the sub-factor structure was preserved.

Response

Thank you for your comment. We determined the sample size using the Kaiser-Meyer-Olkin test. For this analysis, the sample size was sufficient. We added the following sentence and supplemental file 1.

------Revised Manuscript------

P10, Line 190

We also used Kaiser-Meyer-Olkin test to confirm that the sample size was appropriate [21].

P14, Line 252

We also evaluated the measures of sampling adequacy (MSA) using the Kaiser-Meyer-Olkin test for each item, and the results indicate that sample size was sufficient for factor analysis (meritorious:0.87~ marvelous:0.95), as shown in S1 Table.

P26, Line 345 　　　　　　　　　　　　　　　　　　　　　　　　　

Also, the result of sample size by using Kaiser-Meyer-Olkin test was appropriate [21]. 

Comment

#8

Clinical Implications/Research Implication

Page 17, lines 301-302. “standard” should be explained, whether a unit 

falls below the standard will be evaluated according to a written/valid 

procedure or according to the score obtained from the scale!

Response

Thank you for your comment. There are criteria based on the scoring guideline; thus we added the following sentences.

------Revised Manuscript------

P27, Line367

The HWE-AT score guidelines determine that 1.00-2.99 is “Needs Improvement”, 3.00-3.99 is “Good”, and 4.00-5.00 is “Excellent” [15]. If a unit falls below the standard, issues can be identified, and steps can be taken to resolve them. Moreover, AACN had previously surveyed HWE on a five-point scale between 2006 and 2018 [32]. Using the HWE - AT-J, we will be able to evaluate the current state in each unit.

Comment

#9

Figure 1

“Testeing” should be fixed

Response

P7, Line 106

Thank you for your comment. We fixed Fig 1.

Reviewer #2

Comment

#1

Since HWE-AT-j is an evaluation method derived from the original HWE-AT, 

it is necessary to discuss the comparison with the original HWE-AT (distribution of the sample, factor loading for each question, etc.).

Response

Thank you for your useful comment, we discussed the comparison with the original HWE-AT.

Comment

#2

The author started the analysis with confirmatory factor analysis. 

Please explain why you did not consider factor structure, factor loading,etc. in the exploratory factor analysis.

Response

Thank you for your reasonable advice. 

We considered performing the EFA again; however, we decided not to perform this analysis because of the following reasons:

1. The original the HWE-AT was not developed using EFA.

2. The purpose of this study was to assess construct validity and reliability of the HWE-AT-J. Revising the original HWE-AT was not within the purpose of this study. 

3. Thus far, no studies performed EFA on the HWE-AT in any language. Also, we determined that our study’s purpose will be achieved using only CFA.

We added the reasons we did not perform EFA in the method section. 

------Revised Manuscript------

P11, Line201

We did not perform exploratory factor analysis (EFA) because as far as we know the HWE - AT was not developed using EFA [23]. Moreover, the purpose of this study was to consider structural equivalence between the Japanese version and the original version. Therefore, we chose to perform CFA.

Comment

#3

The reader cannot evaluate the ceiling effect and floor effect from 

table2. Please make it a general table (mean, SD).

Response

Thank you for your useful comment. We revised table 2 according to your comment to make it easier to understand for the readers. We deleted the ceiling and floor effects as a part of the result and discussion, and revised Table 4. We also added new sentences.

------Revised Manuscript------

Revised table2

・P18, Line282

These sentences were deleted.

・P17, Line266

The authors checked for ceiling and floor effects; when SD was added or subtracted from the mean value, none of the items exceeded the maximum value of 5 or were below the minimum value of 1, as shown in Table 2. Therefore, the results showed no significant bias in the distribution of responses to each questionnaire item. 

・P26,Line343

 Additionally, there is no ceiling or floor effect; thus, the scale is considered acceptable.

These sentences were added.

・P17, Line270

The authors determined that the ceiling effect was observed when 0% to 14% of respondents rated “5”. The floor effect was determined when 7% to 40% of respondents rated as “0”. We found Q3, Q 4, Q7, Q8, Q10, Q14, Q16, Q17, and Q 18 had floor effects.

・P25, Line329

　Floor effects were observed in the following items (Q3, Q4, Q7, Q8, Q10, Q14, Q16, Q17 and Q18). As consistent with the original scale, a five-point Likert scale was used to measure the items. The result reflects the current situation of Japanese nursing practice that is hierarchical in nature. Therefore, the floor effect does not hamper the generalization of the scale adequacy.

Comment

#4

In table 5, Cronbach's increases when Q10 is deleted. Please explain 

this in the discussion. Is it the same trend in the original paper?

Response

Thank you for your critical comment. 

The original paper[1] does not show internal consistency in detail, and we do not think it is comparable.Our study purpose was to use the same specification method as the original study, and to maintain consistencey with the original HWE-AT study, we did not remove any specific questions. 

1. Connor JA, Ziniel SI, Porter C, Doherty D, Moonan M, Dwyer P, et al. Interprofessional use and validation of the AACN Healthy Work Environment Assessment Tool. Am J Crit Care. 2018;27(5): 363–371. https://doi.org/10.4037/ajcc2018179.

Comment

#5

(line 241) There is no mention of this in table2. Please correct it.

Response

Thank you for your comment. We mentioned Table 2 in text.

------Revised Manuscript------

These sentences were deleted.

・P17, Line275

The mean scores, standard deviations, and factor loadings are presented in Table 2.

These sentences were added.

・P17, Line276

 The mean scores and standard deviations are shown in Table2.

Comment

#6

Please list Χ2 and df in table3. The researcher is free to use any index 

to some extent, but Χ2 is usually necessary.

Response

Thank you for your comment. We added X2 and df values in Table 3.

------Revised Manuscript------

Revised table3

P20, Line286

Comment

#7

What is the reason why the factor loading in table3 is not standardized?

In other papers, standardized factor loading is used.

Response

Thank you for pointing that out. We changed the standardized factor loading as table2.

------Revised Manuscript------

Revised table2

・P18, Line282

Comment

#8

In Japan, what is the difference between a certified nurse and a specialist?　

What kind of license is it? Please describe.

Response

Thank you for asking this question. In Japan, a CN has received at least 600 hours of training in a specific field, and a CNS has completed a master's degree in a specific field.

------Revised Manuscript------

P24, Line323

A Certified nurse (CN) is those who received at least 600 hours of training in a special

field and who have often served as leaders of a unit because of their specialized

knowledge and expertise. Also, a Certified Nurse Specialist (CNS)has completed a

master’s degree in a specific field.

Comment

#9

The percentages for staff and administration in table 4 are blank. 

Please correct it.

Response

 Thank you very much for letting us know. We revised the tables and combined the information in Table 4 with Table 1.

------Revised Manuscript------

Revised table1

・P14, Line261

Comment

#10

Why don't you combine the characteristics in table 1 and 4 into the same 

table? This is a suggestion.

Response

Thank you for your comment. We combined Tables 1 and 4, and renumbered the tables accordingly.

------Revised Manuscript------

Revised table1

・P14, Line 261

#11

Please attach the Japanese version of the questionnaire (HWE) prepared 

by the author to the supporting information.

Response

Thank you for your comment. We attached the Japanese version of HWE as a supplemental file 2.

------Revised Manuscript------

S2 Table 

Reference

Red articles were added after first submission

1. Massoudi DAH, Hamdi DSSA. The consequence of work environment on employees productivity. IOSR JBM. 2017;19(1): 35–42. https://doi.org/10.9790/487X-1901033542.

2.Thing R, et al. WHO Healthy Workplace Framework and Model Synthesis Report; 2010. Available:

https://apps.who.int/iris/bitstream/handle/10665/113144/9789241500241_eng.pdf.

3.McHugh MD, Rochman MF, Sloane DM, Berg RA, Mancini ME, Nadkarni VM, et al. Better nurse staffing and nurse work environments associated with increased survival of in-hospital cardiac arrest patients. Med Care. 2016;54(1): 74–80. https://doi.org/10.1097/MLR.0000000000000456.

4.Olds DM, Aiken LH, Cimiotti JP, Lake ET. Association of nurse work environment and safety climate on patient mortality: A cross-sectional study. Int J Nurs Stud. 2017;74: 155–161. https://doi.org/10.1016/j.ijnurstu.2017.06.004.

5. Kelly L, Todd M. Compassion fatigue and the healthy work environment. AACN Adv. Crit. Care. 2017; 28(4):351–274. https://doi.org/10.4037/aacnacc2017283.

6. Manojlovich, M. De Cicco, B. Healthy work environment, nurse-physician communication, and patients’ outcome. Am Crit Care .2007; 16(6): 536–543.

7. Manojlovich, M, Antonakos, C L, Ronis, D L. Intensive care units, communication between nurses and physicians, and patients' outcomes. Am. J. Crit. Care. 2009; 18(1): 21–30. https://doi.org/10.4037/ajcc2009353

8. Cohen J, Rodgers, Y M. Contributing factors to personal protective equipment s hortages during the COVID-19 pandemic. Prev. Med. 2020; 141. 1–24. https://doi.org/10.1016/j.ypmed.2020.106263

9. Chatzittofis A, Constantinidou A, Artemiadis A, Michailidou K, et al. The role of perceived organizational support in mental health of healthcare workers during the COVID-19 pandemic: A cross-sectional study. Front. Psychiatry. 2021; 12. https://doi.org/1–6. 10.3389/fpsyt.2021.707293.

10. Shah M, Roggenkamp M, Ferrer L, Burger V, et al. Mental health and COVID-19: The psychological implications of a pandemic for nurses. Clin. J. Oncol. Nurs. 2021; 25(1). 69–75. https://doi.org/10.1188/21.CJON.69-75.

11. Sezgin D, Dost A, Esin M N. Experiences and perceptions of Turkish intensive care nurses providing care to Covid-19 patients: A qualitative study. Int. Nurs. Rev. 2021; 1–13. https://doi.org/10.1111/inr.12740.

12.Munro CL, Hope AA. Healthy work environment: Resolutions for 2020. Am J Crit Care. 2020;29(1): 4–6. https://doi.org/10.4037/ajcc2020940.

13. American Association of Critical Care Nurses. AACN Standards for establishing and sustaining healthy work environments: A Journey to excellence. American Association of Critical Care Nurses; 2005. https://doi.org/10.4037/ajcc2005.14.3.187.

14. American Association of Critical Care Nurses. AACN Healthy Work Environment Assessment Tool; 2009. Available: https://www.aacn.org/nursing-excellence/healthy-work-environments/aacn-healthy-work-environment-assessment-tool.

15. American Association of Critical are Nurses. AACN Healthy Work Environment

Assessment. Team Assessment Results. Available: https://www.aacn.org/WD/HWE/Docs/HWESampleAssessment.pdf

16.Huddleston P, Gray J. Measuring nurse leaders’ and direct care nurses’ perceptions of a healthy work environment in an acute care setting, part 1: A pilot study. J Nurs Adm. 2016;46(7–8): 373–378. https://doi.org/10.1097/NNA.0000000000000361.

17. Sousa VD, Rojjanasrirat W. Translation, adaptation and validation of instruments or scales for use in cross-cultural health care research: A clear and user-friendly guideline. J Eval Clin Pract. 2011;17(2): 268–274. https://doi.org/10.1111/j.1365-2753.2010.01434.x. 

18.Yusoff MSB. ABC of content validation and content validity index calculation. Educ Med J. 2019;11(2): 49–54. https://doi.org/10.21315/eimj2019.11.2.6. 

19.Kyriazos TA. Applied psychometrics: Sample size and sample power considerations in factor analysis (EFA, CFA) and SEM in general. Psychology. 2018;9: 2201–2230. https://doi.org/10.4236/psych.2018.98126. 

20.Hoe SL. Issues and procedures in adopting structural equation modeling technique. J Appl Quant Methods. 2008;3: 76–83. Available: 

https://ink.library.smu.edu.sg/sis_research/5168.

21.Kaiser, Rise. Little Jiffy Mark IV. Educational and Psychological Measurement.

1974; 34, 111-117. https://doi.org/10.1177/001316447403400115.

22.Zou GY. Sample size formulas for estimating intraclass correlation coefficients with precision and assurance. Stat Med. 2012;31(29): 3972–3981. https://doi.org/10.1002/sim.5466.

23. Connor JA, Ziniel SI, Porter C, Doherty D, Moonan M, Dwyer P, et al. Interprofessional use and validation of the AACN Healthy Work Environment Assessment Tool. Am J Crit Care. 2018;27(5): 363–371.

24.McDonald RP, Ho MH. Principles and practice in reporting structural equation analyses. Psychol Methods. 2002;7: 64–82. https://doi.org/10.1037/1082-989X.7.1.64.

25.Hu L, Bentler PM. Cutoff criteria for fit indexes in covariance structure analysis: Conventional criteria versus new alternatives. Struct Equ Model. 1999;6(1): 1–55. https://doi.org/10.1080/10705519909540118.

26.Kenny DA, Kaniskan B, McCoach DB. The performance of RMSEA in models with small degrees of freedom. Sociol Methods Res. 2015;44(3): 486–507. https://doi.org/10.1177/0049124114543236.

27. McHorney, C. A., Tarlov, A. R. Individual-patient monitoring

in clinical practice: Are available health status surveys adequate?

Qual Life Res. 1995; 4(4): 293–307. https://doi.org/10.1007/

BF015 93882.

28. Koo TK, Li MY. A guideline of selecting and reporting intraclass correlation 

 coefficients for reliability research. J Chiropr Med. 2016;15(2): 155–163. https://doi.org/10.1016/j.jcm.2016.02.012.

29.Cronbach LJ. Coefficient alpha and the internal structure of tests. Psychometrika. 1951;16: 297–334. Available: https://link.springer.com/content/pdf/10.1007/BF02310555.pdf. DOI: 10.1007/BF02310555.

30. Maskey R, Fei J, Nguyen HO. Use of exploratory factor analysis in maritime research. Asian J Ship Logist. 2018;34(2): 91–111. https://doi.org/10.1016/j.ajsl.2018.06.006.

31. Landis JR, Koch GG. The measurement of observer agreement for categorical data. Biometrics. 1977;33: 159–174. https://doi.org/10.2307/2529310.

32. Beth U, Connie B, Linda C,et al.Crinical care nurse work environment 2018: Findings and implications.2019; 39(2), 67-84. https://doi.org/10.4037/ccn2019605.

mp

---

## [Decision Letter · Decision Letter 1]

25 Apr 2022

Development and Initial Validation of the Japanese Healthy Work Environment Assessment Tool for Critical Care Settings

PONE-D-21-39634R1

Dear Dr. Kitayama,

We’re pleased to inform you that your manuscript has been judged scientifically suitable for publication and will be formally accepted for publication once it meets all outstanding technical requirements.

Kind regards,

Fatih Özden, PhD

Academic Editor

PLOS ONE

Additional Editor Comments (optional):

Reviewers' comments:

Reviewer's Responses to Questions

**Comments to the Author**

1. If the authors have adequately addressed your comments raised in a previous round of review and you feel that this manuscript is now acceptable for publication, you may indicate that here to bypass the “Comments to the Author” section, enter your conflict of interest statement in the “Confidential to Editor” section, and submit your "Accept" recommendation.

Reviewer #1: All comments have been addressed

Reviewer #2: All comments have been addressed

2. Is the manuscript technically sound, and do the data support the conclusions?

Reviewer #1: Yes

Reviewer #2: Yes

3. Has the statistical analysis been performed appropriately and rigorously? 

Reviewer #1: Yes

Reviewer #2: Yes

4. Have the authors made all data underlying the findings in their manuscript fully available?

Reviewer #1: Yes

Reviewer #2: Yes

5. Is the manuscript presented in an intelligible fashion and written in standard English?

Reviewer #1: Yes

Reviewer #2: Yes

6. Review Comments to the Author

Reviewer #1: (No Response)

Reviewer #2: (No Response)

7. PLOS authors have the option to publish the peer review history of their article (what does this mean?). If published, this will include your full peer review and any attached files.

Reviewer #1: No

Reviewer #2: No

---

## [Editor Report · Acceptance letter]

4 May 2022

PONE-D-21-39634R1 

Development and Initial Validation of the Japanese Healthy Work Environment Assessment Tool for Critical Care Settings 

Dear Dr. Kitayama:

I'm pleased to inform you that your manuscript has been deemed suitable for publication in PLOS ONE. Congratulations! Your manuscript is now with our production department. 

Kind regards, 

on behalf of

Dr. Fatih Özden 

Academic Editor

PLOS ONE